# A Phase-Coded FMCW-Based Integrated Sensing and Communication System Design for Maritime Search and Rescue

**DOI:** 10.3390/s25175403

**Published:** 2025-09-01

**Authors:** Delong Xing, Chi Zhang, Yongwei Zhang

**Affiliations:** 1School of Transportation and Civil Engineering, Nantong University, Nantong 226019, China; xdl@stmail.ntu.edu.cn; 2School of Information and Technology, Nantong University, Nantong 226019, China; zhangchi1845@stmail.ntu.edu.cn

**Keywords:** maritime search and rescue (SAR), integrated sensing and communication (ISAC), multiple-input multiple-output (MIMO), phase-coded FMCW (PC-FMCW), compound K-distribution, multitarget detection, Zadoff–Chu sequence

## Abstract

Maritime search and rescue (SAR) demands reliable sensing and communication under sea clutter. Emerging integrated sensing and communication (ISAC) technology provides new opportunities for the development and modernization of maritime radio communication, particularly in relation to search and rescue. This study investigated the dual-function capability of a phase-coded frequency modulated continuous wave (FMCW) system for search and rescue at sea, in particular for life signs detection in the presence of sea clutter. The detection capability of the FMCW system was enhanced by applying phase-modulated codes on chirps, and radar-centric communication function is supported simultaneously. Various phase-coding schemes including Barker, Frank, Zadoff-Chu (ZC), and Costas were assessed by adopting the peak sidelobe level and integrated sidelobe level of the ambiguity function of the established signals. The interplay of sea waves was represented by a compound K-distribution model. A multiple-input multiple-output (MIMO) architecture with the ZC code was adopted to detect multiple objects with a high resolution for micro-Doppler determination by taking advantage of spatial coherence with beamforming. The effectiveness of the proposed method was validated on the 4-transmit, 4-receive (4 × 4) MIMO system with ZC coded FMCW signals. Monte Carlo simulations were carried out incorporating different combinations of targets and user configurations with a wide range of signal-to-noise ratio (SNR) settings. Extensive simulations demonstrated that the mean squared error (MSE) of range estimation remained low across the evaluated SNR setting, while communication performance was comparable to that of a baseline orthogonal frequency-division multiplexing (OFDM)-based system. The high performance demonstrated by the proposed method makes it a suitable maritime search and rescue solution, in particular for vision-restricted situations.

## 1. Introduction

Since the inception of maritime radio communication in the early 20th century, search and rescue (SAR) operations have relied critically on wireless technologies to detect, locate, and assist vessels or individuals in distress. The historical significance of radio communication was first demonstrated during the Titanic disaster, which catalyzed the adoption of the international SOS distress signal at the 1906 Berlin Radiotelegraph Convention under the international telecommunication union (ITU). In the decades that followed, the international maritime organization (IMO) established a series of regulatory frameworks—including the international convention for the safety of life at sea (SOLAS) and the global maritime distress and safety system (GMDSS)—to standardize SAR protocols and modernize distress communication infrastructure. Recent updates to the GMDSS (2011–2022) have advocated for the application of emerging digital and integrated systems to enhance the reliability and responsiveness of SAR under complex maritime conditions [1,2,3]. As shown in Table 1, existing maritime sensing platforms—from spaceborne SAR and ship-borne frequency modulated continuous wave (FMCW) radar to unmanned aerial vehicle (UAV)-based orthogonal frequency-division multiplexing (OFDM)-integrated sensing and communication (ISAC) and electro-optical/infrared imaging—provide a range of capabilities, but suffer from fundamental trade-offs. Spaceborne SAR offers wide area coverage, but is limited by long revisit intervals; conventional marine FMCW radars achieve meter-scale resolution, yet lack Doppler or communication functions; OFDM-based ISAC systems maintain robust links, but deliver coarse ranging under heavy sea clutter; and Electro-Optical and Infrared (EO/IR) payloads deliver sub-meter visual detail only in favorable weather. In Table 1, ΔR denotes the range resolution, or the smallest distance between two targets that each system can distinguish, which largely depends on the signal bandwidth. These gaps motivate the pursuit of a tightly integrated sensing-and-communication paradigm, with sensing as a priority.

Despite decades of evolutionary advances in maritime SAR, traditional systems still treat sensing and communication as separate functions, employing uncoded FMCW or pulsed waveforms with mechanical beam steering or post-processing Doppler filters. Such architectures have achieved limited range resolution, producing high false alarm rates in a substantial sea clutter environment, and they demand bulky apertures or a large transmit power to preserve signal-to-noise ratio (SNR) [8,9,10]. In contrast, emerging ISAC technology adopts unified waveform design and performs detection processing within a single platform [11,12,13], enabling simultaneous multitarget discrimination and micro-Doppler extraction, while maintaining communication function under the maritime conditions.

FMCW radar is widely used in sensing-only applications. However, signals adopted in conventional FMCW radar systems present difficulties for SAR tasks in challenging environments: the probability of detection is low due to the relatively high sidelobe level in the ambiguity function associated with the received FMCW signal. On the other hand, FMCW radar systems are unable to provide communication functions directly. To overcome these constraints, phase-coded FMCW (PC-FMCW) waveforms were then introduced, in which phase modulation is added to the FMCW chirps. The new waveforms enable dual functions out of the FMCW-based platform and a radar-centric ISAC system can be established [14,15].

The benefits of adding phase modulation into FMCW systems are twofold: (1) The phase-modulated system significantly enhances maritime target detection by suppressing sea clutter characterized by compound K-distribution statistics. This model captures both the long-term texture (large-scale wave structure) and short-term speckle (small-scale wave facets) properties of sea surfaces [16,17,18]. Through phase coding, the system effectively mitigates these dual-scale clutter components, improving detection performance in challenging sea conditions [19,20,21]; (2) Communication data can be carried simultaneously by the PC-FMCW signals. As a result, the ISAC system based on PC-FMCW technology can be implemented as a radar-centric solution for maritime SAR operations.

In this work, a MIMO PC-FMCW-based system design was developed and investigated for maritime search and rescue applications. The propagation channel of the sea environment was simulated with a compound K-distribution. A variety of phase-coding schemes for the FMCW chirps, including Barker, Frank, Zadoff–Chu (ZC), and Costas codes, were investigated and their merits for sensing were measured by using the peak sidelobe level ratio (PSLR) and integrated sidelobe level ratio (ISLR) of the autocorrelation function. The communication performance of the PC-FMCW-based radar-centric system is compared with a baseline OFDM system under the same channel condition. The key contributions of this work are summarized as follows:
A comprehensive MIMO FMCW ISAC simulation platform for maritime SAR: We developed a unified simulation platform that integrates realistic maritime clutter modeling, multitarget sensing, and dual-function communication [22,23]. The system incorporated compound K-distributed sea clutter with tunable shape and scale parameters, a flexible MIMO transceiver architecture, and support for four representative phase coding schemes: Barker, Frank, Costas, and Zadoff–Chu [17]. This enables system evaluation from waveform generation and coding to beamforming, range-Doppler processing, and communication decoding, under varying channel and clutter conditions [24,25].Sensing performance analysis under heavy clutter: We evaluated detection metrics such as PSLR, ISLR, and multitarget detection accuracy via permutation-based pairing. Monte Carlo simulations demonstrated the superior autocorrelation and sidelobe suppression of ZC codes [26,27]. In addition, range estimation error is assessed as a function of target range and clutter level, showing that ZC-coded FMCW achieves a higher accuracy than conventional uncoded FMCW, particularly in clutter-dense zones (50–70 m), thus improving robustness in mid-range detection. The system also supports extraction of micro-motion signatures and life-sign features, fulfilling the aim even under strong clutter [28,29,30].Integrated communication analysis in ISAC scenarios: A quadrature phase shift keying (QPSK)-based communication system was used as a base time to evaluate the bit error rate (BER) and channel capacity of the PC-FMCW under the sea clutter environment represented by the compound-K distribution. Results showed that ZC-coded FMCW offers a comparable BER performance over conventional OFDM waveforms across a wide SNR range. This demonstrates the dual-function capability of our ISAC system for both high-resolution sensing and reliable communication, offering a valuable avenue for future maritime SAR platforms operating in degraded environments.

The remainder of this paper is organized as follows. Section 2 presents the radar and clutter models. Section 3 describes the simulation framework, including parameter studies, target detection, and BER evaluation. Section 4 reports the results, and Section 5 concludes the paper.

## 2. System Model

As shown in Figure 1, the system model is established with three principal components: FMCW signal generation with integrated phase coding; MIMO array processing and coherent beamforming; and comprehensive channel modeling that rigorously captures water surface clutter characteristics using a compound-K distribution in both scalar and matrix forms. This section details the mathematical foundations, parameter choices, and theoretical rationale behind each component.

### 2.1. Transmit Radar Signal

The system employed an FMCW waveform for high-resolution range estimation. The instantaneous frequency of a chirp is defined as(1)$$f\left(t\right)=f_{c}+\frac{B}{T_{\mathrm{chirp}}}t,\;0\leq t<T_{\mathrm{chirp}},$$
where $f_{c}$ is the carrier frequency, *B* is the bandwidth, $T_{\mathrm{chirp}}$ is the chirp duration, and the transmitted baseband signal is given by(2)$$s_{\mathrm{tx}}\left(t\right)=\exp\;\left[j2\pi \left(f_{c}\;t+\frac{B}{2T_{\mathrm{chirp}}}t^{2}\right)\right],$$
where the term $f_{c}t$ corresponds to the carrier phase component, while $\frac{B}{2T_{\mathrm{chirp}}}t^{2}$ represents the quadratic phase term induced by the linear frequency modulation. The instantaneous phase is derived from the integral of the instantaneous frequency over time. For a target at range *R*, a round-trip propagation delay τ is introduced, given by(3)$$\tau =\frac{2R}{c},$$
which depends on the target range *R* and the speed of light. The factor of 2 accounts for the signal traveling to the target and back to the radar receiver. This delay yields a beat frequency defined by(4)$$f_{b}=\frac{2R}{c}\cdot \frac{B}{T_{\mathrm{chirp}}},$$
which forms the basis for range estimation. The beat frequency is determined by mixing the transmitted and received signals, which is proportional to both the target range *R* and the chirp rate $B/T_{\mathrm{chirp}}$. This parameter serves as the fundamental basis for range estimation in FMCW radar systems. The range resolution is defined as(5)$$\Delta R=\frac{c}{2B},$$
and the maximum unambiguous range is computed as(6)$$R_{\max}=\Delta R\cdot \frac{N_{s}}{2},$$
where $N_{s}=f_{s}\cdot T_{\mathrm{chirp}}$ is the number of samples per chirp. Range resolution defines the minimum separation required to distinguish two targets. The resolution improves with the bandwidth *B*. This relationship originates from the Fourier transform’s inherent resolution limitations. The maximum unambiguous range is determined by the Nyquist sampling criterion. Here, $N_{s}=f_{s}\cdot T_{\mathrm{chirp}}$ represents the number of samples per chirp, where $f_{s}$ is the sampling frequency. The factor $\frac{1}{2}$ ensures the beat frequency does not exceed half the sampling frequency to prevent aliasing.

### 2.2. Phase Coding Modulation and Autocorrelation Analysis

Each transmitted FMCW chirp from element *p* is modulated by a slow-time phase coding sequence c(n) and expressed as(7)$$s_{\mathrm{tx},p}\left(n,t\right)=c\left(n\right)\cdot s_{\mathrm{FMCW}}\left(t\right),$$
where the baseband FMCW waveform, $s_{\mathrm{FMCW}}\left(t\right)$, is defined in (Equation 2), *n* is the count for the chirps, and c(n) represents the code for phase control, which can be the ZC, Costas, or Frank code. The codes, defined by a pattern, are applied periodically on all the chirps.

Four coding schemes were investigated and their characteristics are as follows.

**Barker Code:** A 13-length binary sequence commonly written as(8)$$c_{\mathrm{Barker}}^{\left(13\right)}=\left[+1,\;+1,\;+1,\;+1,\;\ldots ,\;+1,\;-1,\;+1\right],$$
where the middle elements are omitted for brevity; Barker codes are known for their low autocorrelation sidelobes at short lengths.**Frank Code:** Generated for an integer *M* (e.g., M=4), yielding a sequence of length $M^{2}=16$. The (p,q)-th element is given by(9)$$c_{p,q}=\exp\;\left(j2\pi \;\frac{p\;q}{M}\right),\;0\leq p,q<M,$$
and the final sequence is formed by concatenating rows (or columns) of the resulting matrix.**ZC Code:** A constant-amplitude zero-autocorrelation code of length $N_{\mathrm{zc}}$ defined by(10)$$\begin{array}{cc} z(n)= & \exp\;\left(-j2\pi \;\frac{u\;n(n+1)}{N_{\mathrm{zc}}}\right),\\ \; & \;n=0,1,\ldots ,N_{\mathrm{zc}}-1, \end{array}$$
where *u* is coprime with $N_{\mathrm{zc}}$; this code offers excellent autocorrelation properties.**Costas Code:** Constructed via a permutation-based approach for a given *M*, typically generating a length-$M^{2}$ polyphase sequence. By placing frequency-shifted pulses in a permutation pattern, Costas arrays achieve good randomness and multipath resistance.

An autocorrelation function was adopted to evaluate the performance of the codes, and for the code sequence, c(n), it is defined as(11)$$R\left(l\right)=\sum_{n}c\left(n\right)c^{*}\left(n-l\right),$$
where $c^{*}\left(n-l\right)$ is the conjugate of c(n−l). The characteristics of the phase control codes are examined by comparing the PSLR and ISLR of the autocorrelation function. They can be calculated by(12)$$\mathrm{PSLR}=20\log_{10}\;\left(\frac{A_{\mathrm{main}}}{A_{\max\;\mathrm{sidelobe}}}\right),$$
and(13)$$\mathrm{ISLR}=10\log_{10}\;\left(\frac{\sum_{l\neq 0}{\left|R\left(l\right)\right|}^{2}}{{\left|R\left(0\right)\right|}^{2}}\right),$$
respectively. R(l) denotes the autocorrelation function of the phase-coded signal at the time delay moment, *ℓ*, and R(0) corresponds to the peak of the mainlobe—which is equivalent to $A_{\mathrm{main}}$—and $A_{\max\;\mathrm{sidelobe}}=\max_{l\neq 0}\left|R\left(l\right)\right|$ is the peak sidelobe level. In (Equation 13), $\sum_{l\neq 0}{\left|R\left(l\right)\right|}^{2}$ represents the total energy of all sidelobes (i.e., sum of non-zero-time-delay autocorrelation function), and ${\left|R\left(0\right)\right|}^{2}$ represents the mainlobe energy.

Table 2 provides a concise overview of each coding scheme’s basic characteristics, the length, and the repeating pattern used to fill the 100-chirp CPI. The repetition and truncation operations ensure that every waveform undergoes the same coherent integration period and deposits identical total energy into the matched filter. As a result, any observed differences in peak-to-sidelobe ratio or integrated sidelobe ratio arise exclusively from the intrinsic autocorrelation behavior of the codes rather than from unequal sequence length, processing gain, or time-on-target.

In order to quantify the performance of the codes for phase control, the patterns and period of the codes were optimized and selected using the PSLR and ISLR values as the criterion. As summarized in Table 3, classical FMCW has PSLR = 0.09 dB and ISLR = 18.17 dB. Barker improves these to 1.21 dB and 6.70 dB, while Frank and Costas both reach PSLR = 1.51 dB with ISLRs of 10.04 dB and 13.44 dB, respectively. The ZC sequence outperforms the rest with PSLR = 3.22 dB and ISLR = 1.35 dB, motivating its selection for subsequent simulations.

Figure 2 depicts the normalized autocorrelation of the uncoded FMCW waveform (no coding on chirps) over $\pm (N_{\mathrm{chirps}}-1)$ chirp delays. A perfectly triangular mainlobe of unit amplitude appears at the zero delay, and the value drops with a steady slope along the time delay; at ISLR = 18.17 dB, the ambiguity of detection can be substantial. In Figure 3, the autocorrelation functions of the Barker, Frank, Costas, and ZC phase codes are compared within the same delay range. The peak sidelobe with the Barker code is PSLR = 1.21 dB, and the ISLR is 6.70 dB; with the Frank code, PSLR is 1.51 dB and ISLR is 10.04 dB. The Costas sequence attains the same PSLR (1.51 dB) as the Frank code, but the ISLR (13.44 dB) is slightly higher. In contrast, the ZC sequence produces the best performance, with PSLR = 3.22 dB and ISLR = 1.35 dB; its ambiguity for detection is the smallest. Hence, the ZC code for phase modulation was chosen in the subsequent ISAC study.

### 2.3. MIMO Array Processing and Beamforming

In the proposed collocated MIMO architecture, we employed ULAs for transmit and receive, with the inter-element spacing, d=λ/2, and $\lambda =c/f_{c}$. The pairwise combinations of the $N_{t}=4$ transmit and $N_{r}=4$ receive elements can work as a virtual array of $N_{t}\times N_{r}=16$ elements for receiving, providing enhanced angular resolution and spatial diversity while retaining a single-platform geometry.

On the transmit side, the steering vector for a physical ULA of *N* elements is given by(14)$$a\left(\theta \right)=\left[\begin{array}{c} 1\\ \exp\;\left(-j2\pi \frac{d\sin\theta }{\lambda }\right)\\ \;\;\;\;\;\vdots \\ \exp\;\left(-j2\pi \frac{(N-1)d\sin\theta }{\lambda }\right) \end{array}\right],$$
and it is extended to a virtual array by combining the transmit and receive steering vectors for each Tx–Rx pair. The output from the virtual array after beamforming is represented by(15)$$s_{\mathrm{bf}}\left(n,t\right)=\sum_{m=1}^{16}r\left(n,t,m\right)\;\bar{a_{m}\left(\theta_{t}\right)},$$
where r(n,t,m) denotes the received signal at channel *m* during the *n*-th chirp and $\theta_{t}$ is the assumed target angle.

To generate the transmitted signal for the MIMO array, the transmitted signal, $S\left(n,t\right)\in C^{4\times 1}$, at each element is constructed as(16)$$S\left(n,t\right)=c\left(n\right)\cdot s_{\mathrm{FMCW}}\left(t\right).$$

### 2.4. Clutter and Channel Modeling

To explicitly describe the contribution of each transmit and receive antenna pair to the received signal, we expand the original matrix-form channel model into an element-wise formulation. The received signal from the 4 Rx antennas can be written as(17)Y(n,t)=H(n,t)S(n,t)+W(n,t),
where $H\left(n,t\right)\in C^{4\times 4}$ is the full MIMO channel matrix for time *t*, and W(n,t) is the receiver noise vector. Beamforming is then carried out on the received signals with the virtual array theorem.

We reformulate the channel matrix H(n,t) by separating the contributions of the dynamic human target (torso and arm) and sea clutter scatterers as follows:(18)$$H\left(n,t\right)=\sum_{l=1}^{2}\alpha_{l}\left(n,t\right)\;a_{r}\left(\theta_{l}\right)\;a_{t}^{H}\left(\theta_{l}\right)+\sum_{l=3}^{L}A_{l}\sqrt{T_{l}\left(n\right)}\;S_{l}\left(n,t\right)\;a_{r}\left(\theta_{l}\right)\;a_{t}^{H}\left(\theta_{l}\right).$$
In (Equation 18), the first two paths (l=1,2) model human micro-Doppler effects. The instantaneous ranges of torso and arm motion can be described by(19)$$R_{\mathrm{torso}}\left(n\right)=R_{0}+A_{\mathrm{bod}}\sin\left(2\pi f_{\mathrm{bod}}t_{n}\right),$$(20)$$R_{\mathrm{arm}}\left(n\right)=R_{0}+A_{\mathrm{arm}}\sin\left(2\pi f_{\mathrm{arm}}t_{n}\right),$$
where $R_{0}$ is the nominal distance, $A_{\mathrm{bod}},A_{\mathrm{arm}}$ are displacements at maximum distances, and $f_{\mathrm{bod}}=0.2$ Hz, $f_{\mathrm{arm}}=1.0$ Hz are motion frequencies.

The corresponding Doppler frequencies are then derived as(21)$$f_{d,\mathrm{torso}}\left(n\right)=\frac{2}{\lambda }\frac{dR_{\mathrm{torso}}\left(n\right)}{dt},$$(22)$$f_{d,\mathrm{arm}}\left(n\right)=\frac{2}{\lambda }\frac{dR_{\mathrm{arm}}\left(n\right)}{dt}.$$

These motions are embedded in the multipath propagation model with the parameters(23)$$\alpha_{1}\left(n,t\right)=A_{\mathrm{torso}}\exp\left[j\left(\phi_{1}-2\pi f_{d,\mathrm{torso}}\left(n\right)t\right)\right],$$(24)$$\alpha_{2}\left(n,t\right)=A_{\mathrm{arm}}\exp\left[j\left(\phi_{2}-2\pi f_{d,\mathrm{arm}}\left(n\right)t\right)\right].$$

The remaining L−2 propagation paths (l=3,…,L) were modeled by a compound K-distribution where scatters we placed randomly in the range of 50 to 70 m, and each scatter was assigned a Doppler frequency which was generated from a uniform distribution.

As shown in Figure 4, the composite channel model simultaneously captures the characteristics of the time-varying micro-motions of human targets (torso and arm oscillations) and the clutter of the sea surface represented by the non-Gaussian compound K distribution. By exploiting the advantage of the MIMO, every transmit–receive antenna pair is an independent propagation path, and ultimately a virtual array of $N_{t}\times N_{r}$ elements is formed. The signals from the virtual array were processed subsequently for target detection with micro-Doppler effects in the sea clutter environment.

### 2.5. Received Signal Formulation

At the receiver, the echoes from each transmit antenna and path are sampled by each of the 4 receiving antennas. This results in 16 virtual channels formed by all 4×4 Tx-Rx pairings.

The digital signal collected by the *r*-th receive antenna for the *n*-th chirp and time index *t* is(25)$$y_{r}\left(n,t\right)=\sum_{p=1}^{4}\sum_{l=1}^{L}\alpha_{l}\left(n,t\right)\;a_{r}\left(\theta_{l}\right)\;a_{t,p}\left(\theta_{l}\right)\;s_{p}\left(n,t\right)+w_{r}\left(n,t\right),$$
where $y_{r}\left(n,t\right)$ denotes the baseband signal received at the *r*-th antenna during the *n*-th chirp and fast-time index *t*. The term $s_{p}\left(n,t\right)$ represents the FMCW waveform transmitted by the *p*-th antenna. The complex gain $\alpha_{l}\left(n,t\right)=A_{l}\sqrt{T_{l}\left(n\right)\;S_{l}\left(n,t\right)}e^{j\phi_{l}}$ models the *ℓ*-th propagation path, where $T_{l}\left(n\right)∼\mathrm{Gamma}\left(m,\omega /m\right)$ captures the long-term texture variation and $S_{l}\left(n,t\right)∼\mathrm{CN}\left(0,1\right)$ represents the fast-varying speckle. The vectors $a_{t,p}\left(\theta_{l}\right)$ and $a_{r}\left(\theta_{l}\right)$ are the transmit and receive steering vectors at angle $\theta_{l}$, and $w_{r}\left(n,t\right)∼\mathrm{CN}\left(0,\sigma^{2}\right)$ is additive Gaussian noise.

## 3. Signal Processing for Sensing

This section outlines the complete signal generation and processing pipeline of the proposed 4-transmit, 4-receive (4×4) MIMO PC-FMCW radar system. The simulation encompasses target motion modeling, sea clutter generation, phase coding, beamforming, and signal decomposition steps. To provide a concise end-to-end overview of the simulation pipeline, Algorithm 1 summarizes the main steps.
**Algorithm 1** MIMO Radar System Simulation Workflow**Input:** System parameters: $f_{c}$, *B*, $T_{\mathrm{chirp}}$, PRF, $N_{\mathrm{chirps}}$

**Input:** Target parameters: $R_{0}$, $A_{\mathrm{bod}}$, $f_{\mathrm{bod}}$, $A_{\mathrm{arm}}$, $f_{\mathrm{arm}}$

**Output:** RTI image, Doppler spectrum, motion separation results

 1: **Initialize**:

  $c\leftarrow 3\times 10^{8}$; $\lambda \leftarrow c/f_{c}$

  Generate 4×4 ULA steering vector for angle $\theta_{\mathrm{target}}$

 2: **Generate phase coding sequence:**

  $\mathrm{codeSeq}\leftarrow \mathrm{GenerateCode}(N_{\mathrm{chirps}},\mathrm{codeType})$

 3: **for** each chirp n=1 to $N_{\mathrm{chirps}}$ **do**

 4:  **Target motion model:**

   Compute $R_{\mathrm{torso}}\left(n\right)$ and $R_{\mathrm{arm}}\left(n\right)$

 5:  **Radar return synthesis:**

   Combine torso, arm, and clutter returns

 6:  **Multichannel baseband signal:**

   Modulate by codeSeq and add Additive White Gaussian Noise(AWGN) on each virtual channel

 7: **end for**

 8: **Beamforming:** Sum over all channels using steering vector

 9: **Range processing:** Apply windowed Fast Fourier Transform(FFT) over fast-time

 10: **Clutter suppression:** Estimate and subtract background

 11: **Target detection:** Select bin with maximum $\sigma_{k}/\mu_{k}$

 12: **Motion separation:** Bandpass filter for torso and arm

 13: **Visualization:** Generate spectrogram, and envelopes


Algorithm 1 outlines every major step—from initialization, motion and clutter synthesis, phase coding and reception, through beamforming, range FFT, clutter filtering, and motion separation—of the final visualization modules.

### 3.1. Signal Processing for Detection

The received radar signals undergo a sequential signal processing pipeline as illustrated in Figure 5. The received signals from the 16-channel virtual array are beamformed using steering vectors corresponding to the target angle $\theta_{t}$. The signal after beamforming is then subjected to fast-time FFT for ranging. Prior to transformation, each chirp signal is weighted by a Hann window, defined as(26)$$w_{\mathrm{Hann}}\left(t\right)=0.5\left(1-\cos\left(\frac{2\pi t}{N_{s}-1}\right)\right),\;0\leq t<N_{s},$$
which reduces spectral leakage and suppresses sidelobe interference by smoothly tapering the signal edges. The windowed FFT is performed as(27)$$X\left(n,k\right)=\sum_{t=0}^{N_{s}-1}y_{\mathrm{bf}}\left(n,t\right)\;w_{\mathrm{Hann}}\left(t\right)\;e^{-j\frac{2\pi }{N_{s}}kt},$$
where X(n,k) represents the range profile for chirp *n* at range bin *k*.

To mitigate strong sea clutter, a two-stage filtering process is applied. First, a sliding-mean filter is computed along the slow-time axis and subtracted to eliminate stationary returns. This is followed by the application of a second-order Butterworth high-pass filter with a normalized cutoff $W_{n}=0.001$ designed to attenuate slow-varying components caused by ocean waves.

Target detection is then performed by evaluating the temporal statistics of each range bin over all *N* received chirps, where *N* denotes the total number of chirps in one coherent processing interval. Specifically, the mean amplitude $\mu_{k}$ and standard deviation $\sigma_{k}$ across all chirps n=1,…,N are calculated as(28)$$\begin{array}{cc} \mu_{k} & =\frac{1}{N}\sum_{n=1}^{N}\left|X\left(n,k\right)\right|, \end{array}$$(29)$$\begin{array}{cc} \sigma_{k} & =\sqrt{\frac{1}{N-1}\sum_{n=1}^{N}{\left(\left|X\left(n,k\right)\right|-\mu_{k}\right)}^{2}}. \end{array}$$

To examine the effect of waveform (phase-coded FMCW signals) and channel modeling (K-distribution to represent sea clutter) independently, we adopted the following approach: after processing the clutter effect, the responses for ranging determination were gated by an adaptive threshold, $\mu_{k}$, which is an average of all responses in the slow time domain, and the range of a target is determined by selecting the bin showing a maximum standard deviation $\sigma_{k}$. It is worth noting that for scenarios where the targets are dense and in a more dynamic environment, advanced detection techniques are required.

An adaptive threshold is first applied to $\mu_{k}$ to eliminate range bins with insufficient energy; then, among the remaining bins, the one exhibiting the largest temporal standard deviation $\sigma_{k}$ is selected as the detection result, since true target returns show greater amplitude fluctuations over consecutive chirps than clutter or noise.

Once the target bin is identified, the signal $X(n,k^{*})$ is analyzed in the frequency domain. Bandpass filters are designed to isolate motion components within specific frequency ranges: 0.1–0.5 Hz for torso oscillation and 1.0–2.0 Hz for arm movement. Median filtering is used to smooth the amplitude envelopes of the separated signals. This final stage enables reliable extraction of human micro-motion characteristics embedded within cluttered environments, culminating in accurate range-Doppler visualization and target discrimination.

As shown in Figure 5, the received MIMO returns from the 4×4 virtual array were first beamformed and then a fast-time FFT was applied to produce range-resolved profiles. To suppress sea clutter, we applied a two-stage filter along each range bin’s slow-time history: a moving-average filter removes background, and a second-order high-pass Butterworth filter eliminates residual low-frequency drift. Thereafter, we computed the mean amplitude and variance across slow time for every range cell and employed an adaptive, threshold-based rule to select the bin most likely containing the human micro-Doppler signature.

### 3.2. Data Cube Formation

At the transmit end, each transmitted FMCW chirp is phase-coded with a unique ZC sequence specific to its transmit antenna. These phase-coded chirps are reflected by targets and clutter, and the corresponding echoes are captured by the $N_{r}\times N_{t}$ ULA. The received data were ensembled into a three-dimensional (3D) data cube with dimensions including fast time, slow time, and spatial channels (Figure 6).

The structured 3D data cube can be mathematically denoted as(30)$$Y\in C^{N_{r}\times N_{\mathrm{chirps}}\times N_{\mathrm{samples}}}$$
where $N_{r}$ is the number of receiving antennas, $N_{\mathrm{chirps}}$ is the number of chirps, and $N_{\mathrm{samples}}$ is the number of fast-time samples per chirp.

### 3.3. Multitarget Detection Simulation

To evaluate the proposed radar system’s ability to resolve multiple targets under realistic maritime clutter, we implement a simulation module that supports both phase-coded FMCW (ZC) and standard FMCW signal models. Six targets are fixed at(31)$$R_{\mathrm{true}}=\left\{20,\;30,\;40,\;50,\;60,\;70\right\}\;m,$$
and embedded within a sea clutter environment containing $N_{\mathrm{clutter}}=100$ scatterers. Each scatterer’s range varies over time as(32)$$R_{\mathrm{cl}}^{\left(i\right)}\left(t\right)=R_{0}^{\left(i\right)}+\delta R^{\left(i\right)}\sin\left(2\pi f_{\mathrm{cl}}^{\left(i\right)}t+\phi_{0}^{\left(i\right)}\right),$$
and its reflection amplitude is modeled as(33)$$\alpha_{\mathrm{cl}}^{\left(i\right)}\left(t\right)=A^{\left(i\right)}\sqrt{T^{\left(i\right)}\left(t\right)}\;S^{\left(i\right)}\left(t\right),$$
where $T^{\left(i\right)}∼\Gamma \left(m,\omega /m\right)$ and $S^{\left(i\right)}∼\mathrm{CN}\left(0,1\right)$, with m=3 and ω=1.

At the receive end, echo from each target is associated with a fixed beat frequency characterized by(34)$$f_{b}^{\left(k\right)}=\frac{2R_{k}}{c}\;\frac{B}{T_{\mathrm{chirp}}},$$
where *B* is the bandwidth and $T_{\mathrm{chirp}}$ the chirp duration. In the standard FMCW mode, beat signals are obtained via simulated dechirping of reflected chirps from both target and clutter paths.

The system employed a 4-transmit, 4-receive ULA MIMO array. Beamforming was applied using the steering vector a(θ). After beamforming, fast-time signals were processed with Hann-windows first, and then FFT was applied to yield range bins. Energy across 512 chirps was integrated to obtain high-SNR for detection procedures. The accumulated energy profile yields six distinct peaks at the true target ranges {20,30,40,50,60,70}m, with the ZC phase-coded FMCW waveform showing a higher peak-to-sidelobe ratio than the standard FMCW mode.

To remove low-frequency clutter bias, a moving-average baseline (window size 20 samples) was subtracted from the energy profile. Range bin candidates were then selected where the residual exceeded(35)Threshold=μ+2σ,
with μ and σ representing the mean and standard deviation of the residual. Peaks were constrained to be at least 5 m apart; the bins with highest energy represented possible targets.

Detected ranges $\hat{R}$ were matched to the ground truth by minimizing total squared error over all permutations. Instead of computing a single-run MSE, we performed $N_{\mathrm{MC}}=1000$ independent Monte Carlo trials, each with fresh noise and clutter realizations. In trial *i*, detection yields an ordered estimate ${\hat{R}}^{\left(i\right)}=\left\{{\hat{R}}_{\pi_{i}\left(1\right)}^{\left(i\right)},\ldots ,{\hat{R}}_{\pi_{i}\left(M\right)}^{\left(i\right)}\right\}$ that minimizes the squared-error cost. The per-trial squared errors are(36)$$e_{k}^{\left(i\right)}={\left({\hat{R}}_{\pi_{i}\left(k\right)}^{\left(i\right)}-R_{k}\right)}^{2},\;k=1,\ldots ,M.$$

The Monte Carlo mean squared error is then(37)$${\mathrm{MSE}}_{\mathrm{MC}}=\frac{1}{N_{\mathrm{MC}}\;M}\sum_{i=1}^{N_{\mathrm{MC}}}\sum_{k=1}^{M}e_{k}^{\left(i\right)}.$$

The above expressions form the theoretical basis for subsequent performance evaluation; numerical results are presented in Section 4.

### 3.4. Communication Performance Comparison

We performed Monte Carlo simulations to assess the communication performance of the ZC-coded FMCW waveform versus a conventional multi-carrier OFDM scheme under a compound K-distributed fading channel. For each SNR value, we averaged over $N_{\mathrm{MC}}=1000$ independent trials, each transmitting $10^{5}$ random bits using 4-QAM modulation.

In each ZC-FMCW trial, a 4-QAM symbol multiplied a normalized Zadoff–Chu sequence of length$$N_{s}=f_{s}\times T_{\mathrm{chirp}},$$
and is transmitted over one chirp interval $T_{\mathrm{chirp}}$ at sampling rate $f_{s}$. The OFDM baseline groups symbols into $\left\lceil 10^{5}/512\right\rceil$ blocks of 512 subcarriers, each IFFT-modulated with a 128-sample cyclic prefix (25% of FFT size).

To emulate heavy-tailed maritime fading conditions, the small-scale fading coefficient is modeled using a compound K-distributed process:(38)$$h^{\left(t\right)}=\sqrt{g^{\left(t\right)}}\cdot r^{\left(t\right)},\;g^{\left(t\right)}∼\mathrm{Gamma}\left(m,\omega /m\right),\;r^{\left(t\right)}∼\mathrm{CN}\left(0,1\right),$$
where m=3 and ω=1 control the shape and scale of the underlying Gamma distribution. The estimated channel is subject to additive Gaussian errors:(39)$${\hat{h}}^{\left(t\right)}=h^{\left(t\right)}+\Delta h^{\left(t\right)},\;\Delta h^{\left(t\right)}∼\mathrm{CN}\left(0,\sigma_{\mathrm{est}}^{2}\right),$$
with $\sigma_{\mathrm{est}}=0.05$, accounting for pilot limitation and receiver mismatch. The corresponding noise variance is defined as(40)$$\sigma^{2}=\frac{1}{2\;{\mathrm{SNR}}_{\mathrm{lin}}},\;{\mathrm{SNR}}_{\mathrm{lin}}=10^{{\mathrm{SNR}}_{\mathrm{dB}}/10},$$
ensuring the per-dimension noise power is $\sigma^{2}$, consistent with the per-symbol SNR. We sweep SNR_dB_ from –5 dB to 25 dB in 2 dB steps.

The receiver applies MMSE equalization:(41)$$\begin{array}{cccc} y_{\mathrm{eq}}^{\left(t\right)} & =\frac{{\left({\hat{h}}^{\left(t\right)}\right)}^{*}}{|{\hat{h}}^{\left(t\right)}{|}^{2}+\sigma^{2}}\;y^{\left(t\right)}, & \; & \;\mathrm{ZC}\text{-}\mathrm{FMCW}, \end{array}$$(42)$$\begin{array}{cccc} Y_{\mathrm{eq}}^{\left(t\right)}\left[k\right] & =\frac{{\left({\hat{H}}^{\left(t\right)}\left[k\right]\right)}^{*}}{|{\hat{H}}^{\left(t\right)}{\left[k\right]|}^{2}+\sigma^{2}}\;Y^{\left(t\right)}\left[k\right], & \; & \mathrm{OFDM}. \end{array}$$
In these MMSE equalizers, ${\hat{h}}^{\left(t\right)}$ and ${\hat{H}}^{\left(t\right)}\left[k\right]$ are the estimated channel gains for the ZC-FMCW chirp and each OFDM subcarrier, respectively. The numerators apply the complex conjugate of the estimated channel to coherently combine signal components, while the denominators normalize by signal and noise energy. This approach minimizes the mean-square error between transmitted and received symbols.

Each equalized stream is demodulated and compared to the original bits. The instantaneous bit error rate for trial *t* is denoted by ${\mathrm{BER}}_{t}$, and the final BER for each SNR point is computed as(43)$$\mathrm{BER}=\frac{1}{N_{\mathrm{MC}}}\sum_{t=1}^{N_{\mathrm{MC}}}{\mathrm{BER}}_{t}.$$

The above expressions form the theoretical basis for subsequent performance evaluation; numerical results are presented in Section 4.

## 4. Experimental Results

This section presents simulation results from the proposed 4-transmit, 4-receive (4×4) MIMO PC-FMCW radar system under realistic maritime conditions. We divided the results into three parts: (1) single-target detection in range and Doppler domains, (2) motion component separation, and (3) Multitarget detection and accuracy evaluation.

The key radar and system parameters are summarized in Table 4.

### 4.1. Single-Target Detection: Range and Doppler

As shown in Figure 7, the mean amplitude profile over all chirps (blue) exceeds the detection threshold (red dashed line) at the true target range of approximately 50 m, flagging this bin as a candidate peak. In parallel, the slow-time standard deviation reaches its maximum of about 12 at 50.10 m (indicated by the red circle and dashed grid lines), which exactly matches the actual target position. The close alignment of peaks in both metrics confirms that combining mean amplitude and temporal variance provides robust and accurate range localization.

As shown in Figure 8, the normalized Doppler spectra of the torso (blue) and arm (red) micro-motion components were obtained by applying a 256-point Hann-windowed FFT to the slow-time envelopes after band-pass filtering in the 0.1–0.5 Hz (torso) and 1.0–2.0 Hz (arm) bands, respectively. Clear peaks appear at approximately 0.2 Hz for the torso and 1.0 Hz for the arm, showing a good agreement with the underlying vital-sign assumptions in the simulation.

### 4.2. Motion Component Separation

Figure 9 plots the normalized, median-filtered amplitude envelopes of the separated torso (top) and arm (bottom) micro-motions over a 10 s slow-time interval. Each slow-time signal was band-pass filtered using a fourth-order, zero-phase IIR filter (0.1–0.5 Hz for the torso; 1.0–2.0 Hz for the arm), processed via the Hilbert transform to extract its analytic envelope, median-filtered (length M=20 samples) to remove sudden peaks, and finally normalized to its peak amplitude. The torso envelope exhibits smooth breathing oscillations (period ∼5 s, 0.2 Hz) with small amplitude fluctuations, whereas the arm envelope reveals rapid arm peaks near 1 Hz.

### 4.3. Multitarget Detection and Accuracy

To quantify the estimation accuracy of each target range, we employed the Monte Carlo-averaged mean squared error ${\mathrm{MSE}}_{\mathrm{MC}}$ defined above. For each distance $R_{k}\in \left\{20,30,40,50,60,70\right\}\;m$, ${\mathrm{MSE}}_{\mathrm{MC}}$ is computed over $N_{\mathrm{MC}}=1000$ independent trials with fresh noise and clutter realizations. This metric provides a direct, statistically robust comparison of the standard FMCW and ZC phase-coded FMCW waveforms under identical simulation settings.

Figure 10 presents the Monte Carlo-averaged squared range estimation error as a function of distance with both waveforms. Across all tested distances from 20 m to 70 m, the ZC coded FMCW waveform consistently yields significantly smaller error than the standard FMCW mode. Notably, a substantial error peak occurs at 60 m based on the conventional FMCW radar, where the average squared error exceeds 2. This corresponds to the region with the strongest compound-K clutter, which was deliberately introduced between 50 m and 70 m as mentioned earlier. In contrast, ZC coding maintains below 0.25 squared error across the entire range, highlighting its robustness against sidelobe leakage and clutter-induced bias. Overall, the ZC waveform stabilizes range estimation under severe clutter, especially in multitarget scenarios with mid-range interference.

### 4.4. Communication Performance Results

Figure 11 illustrates the simulated communication performance of the proposed ZC-coded FMCW waveform (blue circles) compared to a conventional multi-carrier OFDM scheme (red squares) under the complex fading channel. Across the entire SNR range from –5 dB to 25 dB, ZC-FMCW exhibits BER performance similar to OFDM, despite originating from a radar configuration. At moderate-to-high SNRs (≥15 dB), both schemes achieve BERs below $10^{-2}$, and the maximum observed performance gap remains within $2\times 10^{-3}$ even at the highest SNRs. These results confirmed that ZC-FMCW, exhibiting high-performance sensing capability, produced comparable communication reliability compared to OFDM, validating its suitability as a dual-function ISAC waveform in cluttered or multipath-rich environments.

## 5. Discussion

This paper proposed a PC-FMCW based MIMO ISAC system for maritime SAR. In this ISAC design, a compound K-distribution based channel model replicates non-Gaussian, heavy-tailed sea clutter concentrated in the 50–70 m range, enabling realistic performance evaluation through Monte Carlo and ablation studies; four codes (Barker, Frank, Costas, and Zadoff–Chu) were evaluated, with the ZC code achieving the optimal autocorrelation PSLR and ISLR for sidelobe suppression and micro-Doppler fidelity; coherent beamforming on a 4-transmit, 4-receive (4×4) array using steering vectors and a Hann-windowed FFT minimized spectral leakage and sidelobe artifacts in the ISAC receiver; an energy-based peak detector with permutation-based pairing accurately resolved six closely spaced targets (20–70 m), with the ZC coding demonstrating a stable sub-meter range error even under mid-range clutter, the range MSE analysis further confirmed that while standard FMCW suffers bias around 60 m due to clutter, ZC remains robust throughout; and ISAC BER benchmarking under compound clutter showed ZC-coded FMCW is comparable to OFDM across all SNRs, confirming the dual-function potential of the ISAC system for simultaneous sensing and reliable communication in more challenging maritime environment. The ZC coding scheme was adopted for its constant-modulus nature and robust autocorrelation feature for detection; nevertheless, the recent Flag sequence set [31] is a promising alternative and will be investigated in future work within our phase-coded FMCW ISAC pipeline under compound-K sea clutter and embedding realistic Doppler spreads.

In this study, OFDM is used solely as the benchmark for evaluating communication performance as it is most widely used; the advantage of the PC-FMCW based system in comparison to the orthogonal time frequency space (OTFS) and affine frequency division multiplexing (AFDM) systems, lately adopted to mitigate challenges such as doubly-dispersive channels, is to be investigated [32].

*A.* 
*Monte Carlo Evaluation of Detection Probability*


To evaluate system robustness, we performed Monte Carlo simulations. Each trial added a known target signal to AWGN, applied matched filtering, normalized the resulting detection statistic by the estimated noise standard deviation, and compared it against a fixed threshold of γ=3.

Let the received signal in the *i*-th trial be(44)$$r_{i}\left[n\right]=\sqrt{T_{i}S_{i}}\cdot s\left[n\right]+w_{i}\left[n\right],$$
where $T_{i}∼\mathrm{Gamma}\left(m,\omega /m\right)$, $S_{i}∼\mathrm{CN}\left(0,1\right)$, $s\left[n\right]=1/\sqrt{N_{s}}$, and $w_{i}\left[n\right]$ is AWGN. The matched filter output is(45)$$y_{i}=\sum_{n=0}^{N_{s}-1}r_{i}\left[n\right]s^{*}\left[n\right],\;{\mathrm{stat}}_{i}=\frac{|y_{i}|}{\sigma }.$$
Detection probability is computed as(46)$$P_{d}=\frac{1}{N_{\mathrm{trials}}}\sum_{i=1}^{N_{\mathrm{trials}}}I\left({\mathrm{stat}}_{i}>\gamma \right).$$

As shown in Figure 12, the detection probability $P_{d}$ increases steadily with SNR. The system achieves $P_{d}>90\%$ for SNRs above 20 dB, demonstrating robust detection capability under moderate to high noise conditions.

*B.* 
*Effect of Channel-Estimation Accuracy*


To complement the nominal-CSI results, we briefly examine the robustness of the communication link to channel-estimation errors, an unavoidable impairment in maritime ISAC due to motion and heavy-tailed clutter. To isolate this factor, the waveform, modulation, and equalizer settings are fixed; only the estimation perturbation level σ varied. Figure 13 summarizes a BER–SNR sensitivity sweep to the channel-estimation perturbation σ under the compound-K fading (shape m=3, scale ω=1). We use QPSK and the same MMSE equalizer for both ZC-FMCW and OFDM, and σ∈{0.02,0.05,0.08,0.12}, as for each σ the two waveforms shared identical bit/symbol realizations to ensure a fair comparison with Monte Carlo simulations. As seen in Figure 13, BER degrades monotonically as σ increases and a higher SNR floor appears, consistent with the post-equalization SNR loss due to gain/phase mismatch; the effect is accentuated by heavy-tailed compound-K fading. ZC-FMCW exhibits a slightly earlier floor at large σ owing to despreading sensitivity, whereas OFDM distributes estimation noise across subcarriers; the overall trend is the same.

*C.* 
*Resilience to DoA Mismatch*


To assess the impact of beamforming angle errors, we decouple the true direction-of-arrival (DoA) used in data generation from the assumed steering vector used for beamforming and a parameter representing the angle mismatch, $\Delta \theta \in [-5^{\circ },5^{\circ }]$ for the 4×4 virtual ULA with half-wavelength element spacing. For each Δθ, we report the simulated (post-processing) pointing loss after the full chain (range FFT, slow-time clutter suppression, and range-bin selection) as the relative energy drop at the target range bin. The analytical pointing-loss curve $L\left(\Delta \theta \right)=|a_{\mathrm{est}}^{H}a_{\mathrm{true}}{|}^{2}/M^{2}$ closely matches the simulated one over the sweep, with pointwise deviations not exceeding ≈0.07dB, as observed in Figure 14. The loss increases monotonically with |Δθ| and remains small for modest errors: approximately 0.03 dB at $\pm 1^{\circ }$, 0.22 dB at $\pm 3^{\circ }$, and 0.60 dB at $\pm 5^{\circ }$ around a $30^{\circ }$ incidence. This behavior is consistent with the relatively wide mainlobe of the current virtual aperture (effective D≈3λ). In practice, a narrow beam-scan refinement around the nominal angle or diagonal-loaded MVDR (Minimum Variance Distortionless Response) weights can further improve robustness without altering the conclusions of this work.

*D.* 
*Impact of Clutter Parameters*


We performed a parameter sensitivity study to assess the effects of the shape (*m*) and scale (ω) of the compound K-distribution on detection probability. The simulation covers a broad range of clutter statistics, with(47)m∈{1,2,3,4,5,6,7},ω∈{1.0,1.5,2.0,2.5,3.0}.
For each (m,ω) pair, the detection probability $P_{d}$ is estimated via Monte Carlo trials at an SNR of 10 dB. This setup facilitates a comprehensive assessment of system performance across diverse clutter environments, spanning heavy-tailed (low *m*) to near-Gaussian (high *m*) regimes and low to high scale parameters.

Figure 15 illustrates the detection probability $P_{d}$ at an SNR of 10 dB as a joint function of the compound-K distribution’s texture parameter *m* and scale parameter ω, with both parameters selected in a higher range to reflect robust system conditions. The results show that as *m* increases from 2 to 7, $P_{d}$ rises steeply and saturates near unity for (*m* ≥ 5), reflecting the system’s strong resilience against sea clutter with more homogeneous, Gaussian-like texture. Similarly, for a given *m*, increasing the scale parameter ω from 1.0 to 3.0 also yields a significant improvement in detection probability, approaching 1.0 when both *m* and ω are high. This indicates that the system is highly robust under moderate to weak compound-K clutter, and that multi-chirp accumulation can effectively suppress performance degradation even when the scale parameter increases. The optimal region is observed in the upper right corner (*m* ≥ 5, ω≥ 2.0), where $P_{d}$ consistently exceeds 0.95. This highlights the practical advantage of operating in environments with less heavy-tailed clutter statistics or under adaptive clutter estimation.

In summary, this study considered a unified ISAC framework tailored for maritime SAR scenarios. The demonstrated performance across detection accuracy, target localization, and BER validates its suitability as both a research testbed and a reference design tool for future maritime radar systems.

However, several limitations remain. First, the current simulation assumes ideal synchronization and does not incorporate hardware impairments such as phase noise, quantization effects, or mutual coupling between antenna elements. Second, although clutter is modeled with a compound K-distribution, dynamic sea state transitions and non-stationary clutter characteristics are not yet fully captured. In addition, only planar MIMO arrays and static target layouts are considered. Future work will explore real-time processing implementations, adaptive waveform scheduling, dynamic target tracking across multiple frames, and the extension to three-dimensional MIMO array geometries. Enhancing the system’s adaptability to diverse sea conditions and validating it with experimental data will further strengthen its practical utility in maritime SAR missions.

## 6. Conclusions

We have presented an ISAC framework with 4-transmit, 4-receive (4×4) MIMO FMCW radar centric configuration for maritime SAR operations under heavy, non-Gaussian sea clutter. This ISAC platform integrates a compound K-distribution clutter model, high-resolution FMCW waveform synthesis, and four representative phase-coding schemes (Barker, Frank, ZC, Costas) within a coherent beamforming architecture. Through extensive Monte Carlo experiments spanning SNRs from –20 dB to 20 dB, our ISAC framework achieved sub-meter range accuracy when resolving up to six closely spaced targets (20–70 m) and investigated the relationships between the K-distribution parameters, *m* and ω, and detection performance. In parallel, BER comparisons demonstrated that the ZC-coded ISAC waveform matches a conventional OFDM system for communication in channels with sea clutter. These results confirmed the dual-functional capability of the proposed ISAC design for simultaneous sensing and communication, offering a unique advantage in both target detection and life-sign monitoring, and can be a potential solution for maritime radar systems. Future work will extend adaptive clutter modeling, real-time multitarget tracking, and three-dimensional MIMO array configurations to further enhance the operational robustness and field deployability of ISAC systems.

## Figures and Tables

**Figure 1 sensors-25-05403-f001:**
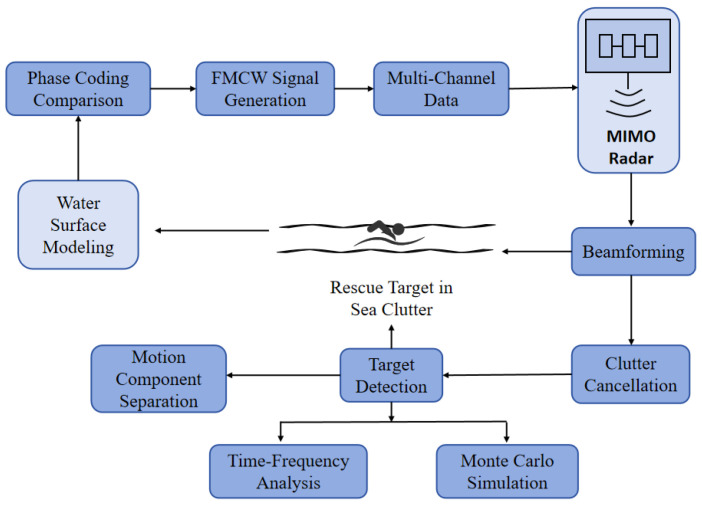
Block diagram of the proposed phase-coded MIMO FMCW ISAC system. Phase-coding comparison selects an optimal waveform (Barker, Frank, Costas, or ZC), which is used to generate multi-channel FMCW signals transmitted by a 4-transmit, 4-receive (4 × 4) MIMO array. Received echoes are coherently beamformed and processed with the clutter cancellation technique, then passed to a target detection stage with classification on motions; time-frequency analysis and Monte Carlo modules, respectively, extract micro-Doppler signatures and quantify detection performance.

**Figure 2 sensors-25-05403-f002:**
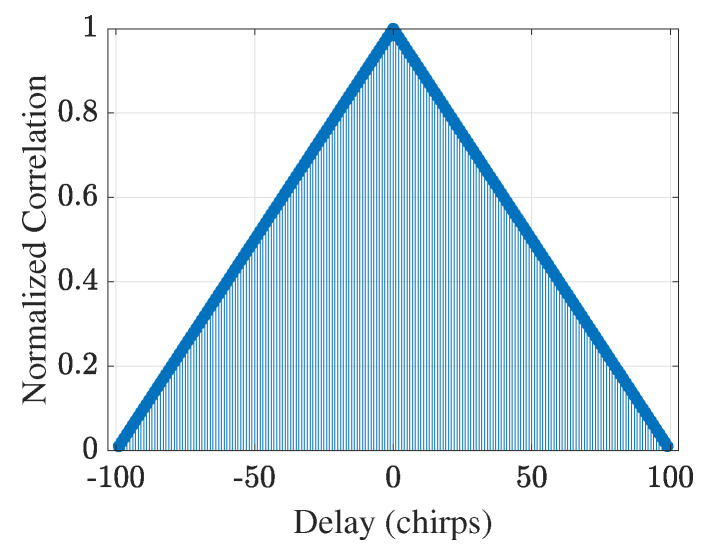
Normalized autocorrelation of the uncoded FMCW waveform over $\pm (N_{\mathrm{chirps}}-1)$ chirp delays, showing a triangular mainlobe and uniformly high sidelobes, which result in poor sidelobe suppression (PSLR ≈0.09 dB) and motivate the use of phase coding for sidelobe reduction.

**Figure 3 sensors-25-05403-f003:**
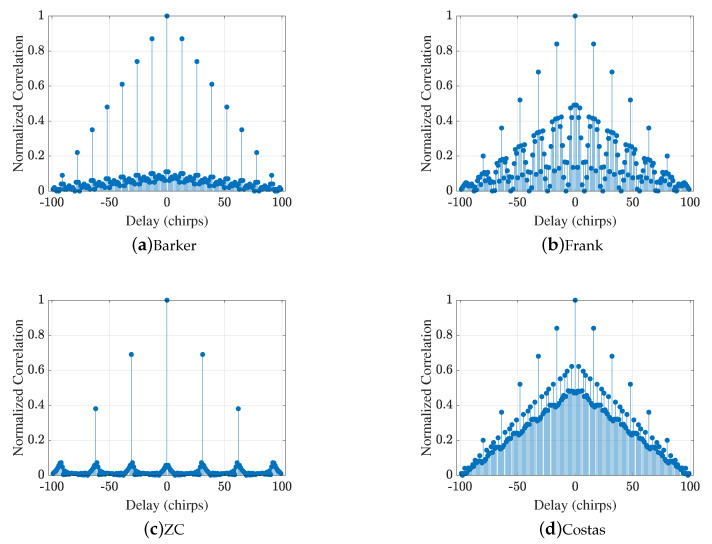
Autocorrelation functions of four phase coding schemes over $\pm (N_{\mathrm{chirps}}-1)$ chirp delays: (**a**) Barker, (**b**) Frank, (**c**) ZC, and (**d**) Costas.

**Figure 4 sensors-25-05403-f004:**
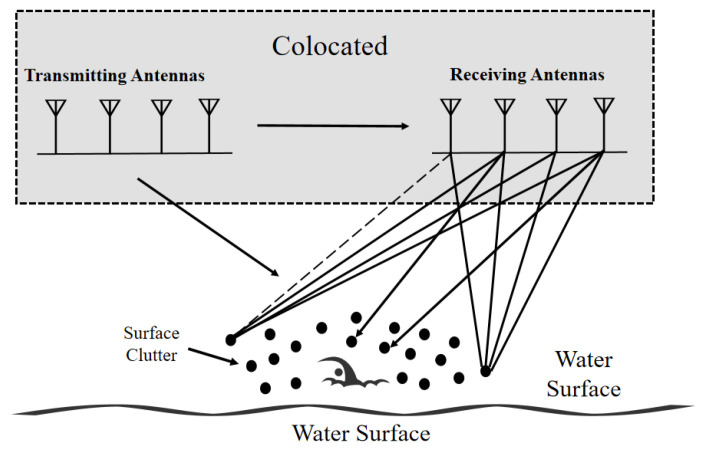
Sea clutter modeling for a collocated 4×4 MIMO FMCW radar. Each of the four transmit antennas illuminates the water surface, producing returns from randomly distributed scatterers (black dots) characterized by a compound K-distribution. Solid lines trace clutter echoes from individual scatterers to each of the four receive antennas. The combined transmit–receive pairs synthesize a 16-element virtual array.

**Figure 5 sensors-25-05403-f005:**
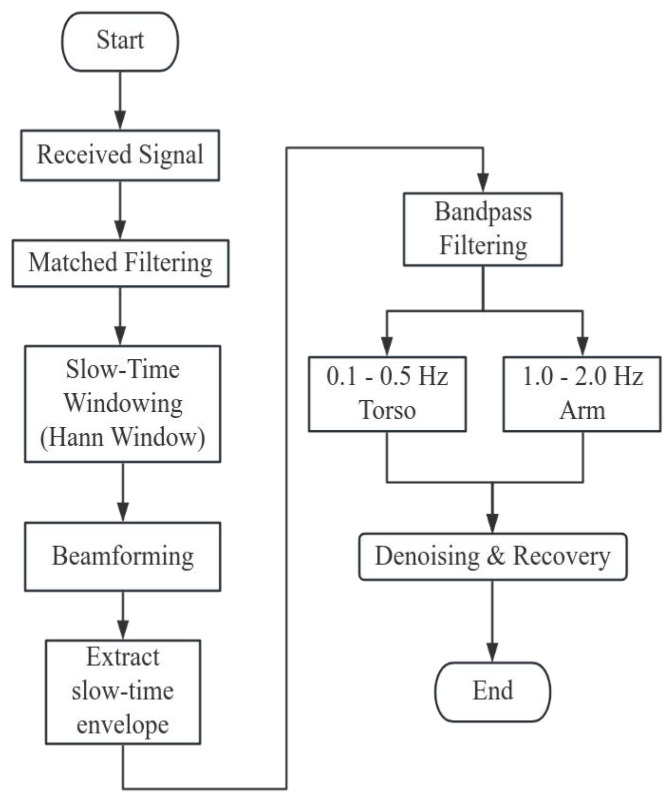
Detailed signal-processing flowchart for vital-sign detection. The received beamformed FMCW returns first undergo matched filtering and slow-time Hann windowing, then MIMO beamforming to yield a range-resolved slow-time envelope. This envelope is extracted and fed into two parallel bandpass filters which are 0.1–0.5 Hz for torso motion and 1.0–2.0 Hz for arm motion—followed by denoising and phase recovery to separate the vital-sign components.

**Figure 6 sensors-25-05403-f006:**
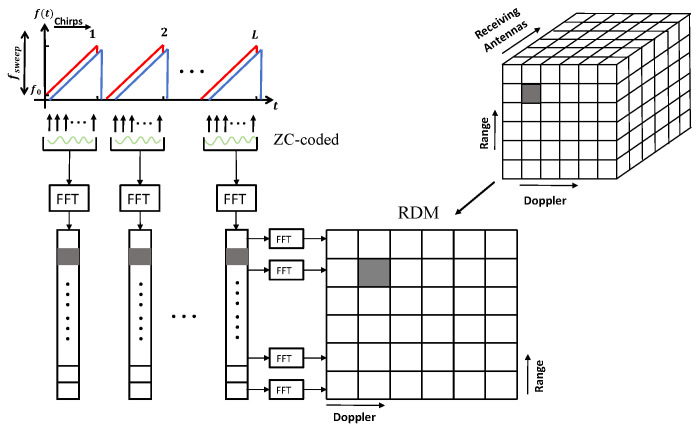
3D sensing data-cube structure for the proposed MIMO FMCW ISAC system. Fast-time FFT on each ZC-coded chirp produces range profiles per virtual channel, which are stacked over slow time and across the 4×4 receive array to form a three-dimensional data cube (range–Doppler–antenna). Subsequent Doppler FFT along the slow-time axis yields the range-Doppler map (RDM) at each virtual element.

**Figure 7 sensors-25-05403-f007:**
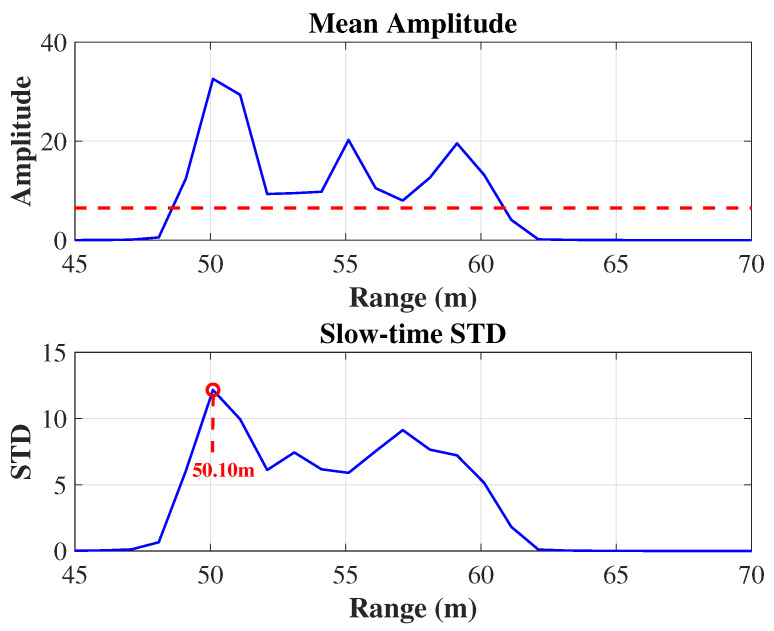
Target detection results across range bins. The top plot shows the mean amplitude profile (blue) with the detection threshold (red dashed line) used to select candidate ranges, while the bottom plot displays the slow-time standard deviation, with the final estimated target range (50.10 m) highlighted by the red marker and dashed grid lines.

**Figure 8 sensors-25-05403-f008:**
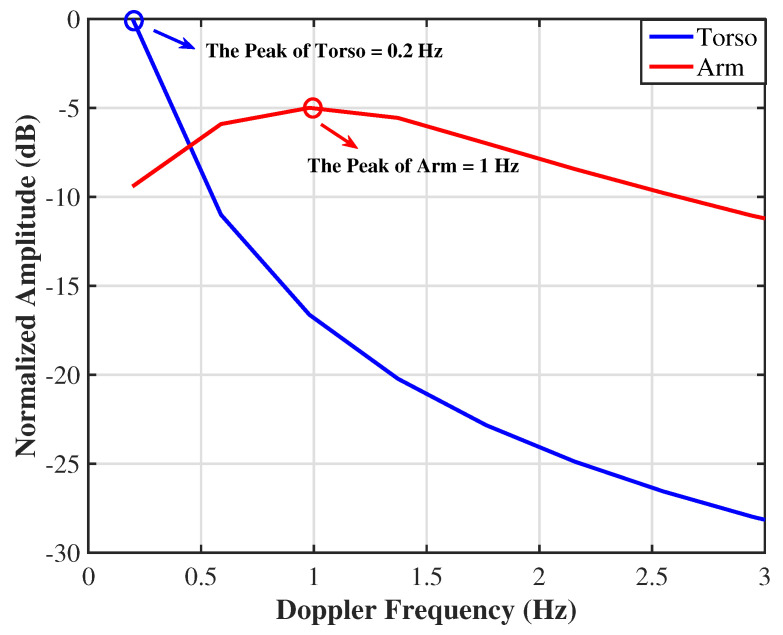
Normalized Doppler spectra of the torso (blue) and arm (red) micro–motion components. Each spectrum was obtained by applying a 256-point Hann-windowed FFT to the slow-time envelopes filtered in the 0.1–0.5 Hz (torso) and 1.0–2.0 Hz (arm) bands, respectively. Clear peaks appear at approximately 0.2 Hz (torso) and 1.0 Hz (arm), corresponding to the underlying vital-sign oscillations.

**Figure 9 sensors-25-05403-f009:**
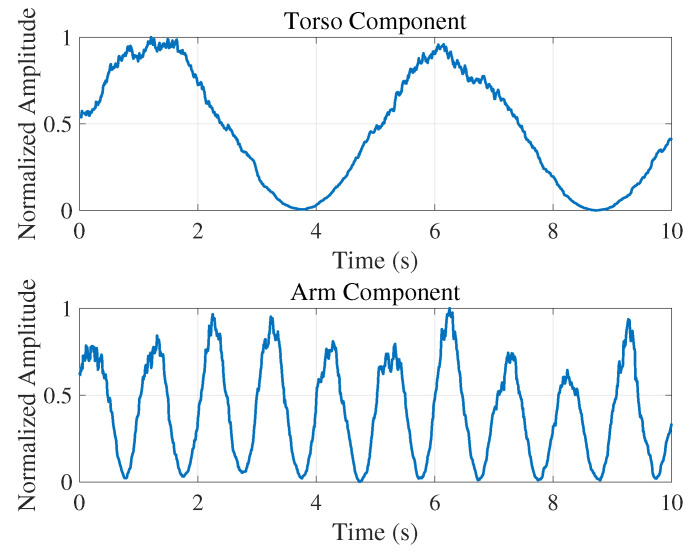
Normalized amplitude envelopes of the separated micro-motion components over slow time (0–10 s). The (**top**) plot shows the torso vibration envelope, extracted by a fourth-order, zero-phase 0.1–0.5 Hz band-pass filter, illustrating smooth breathing cycles at ∼0.2 Hz with realistic low-amplitude fluctuations to reflect measurement noise. The (**bottom**) plot shows the arm motion envelope, extracted by a 1.0–2.0 Hz band-pass filter, revealing rapid hand-motion peaks near 1 Hz with realistic low-amplitude fluctuations. Both envelopes were then median-filtered to remove sudden peaks and normalized to their respective maxima.

**Figure 10 sensors-25-05403-f010:**
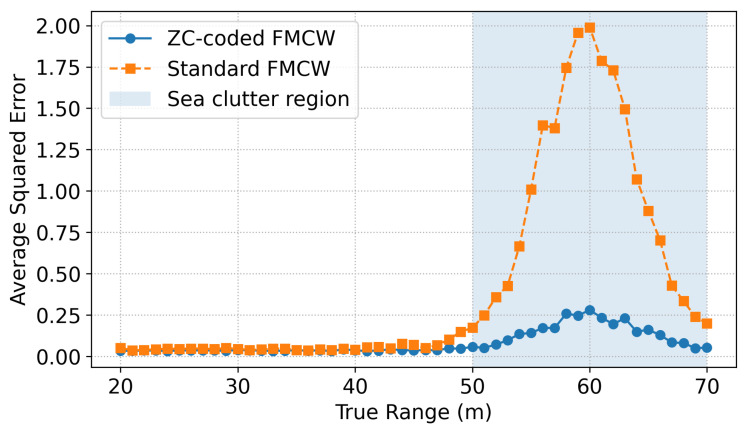
Range-dependent average squared error over 1000 Monte Carlo trials for ZC–coded FMCW (blue circles) and standard FMCW (orange squares). The uncoded FMCW shows a pronounced error peak at 60 m due to clutter concentrated in the 50–70 m region, whereas the ZC-coded waveform maintains sub-0.25 squared error across all ranges.

**Figure 11 sensors-25-05403-f011:**
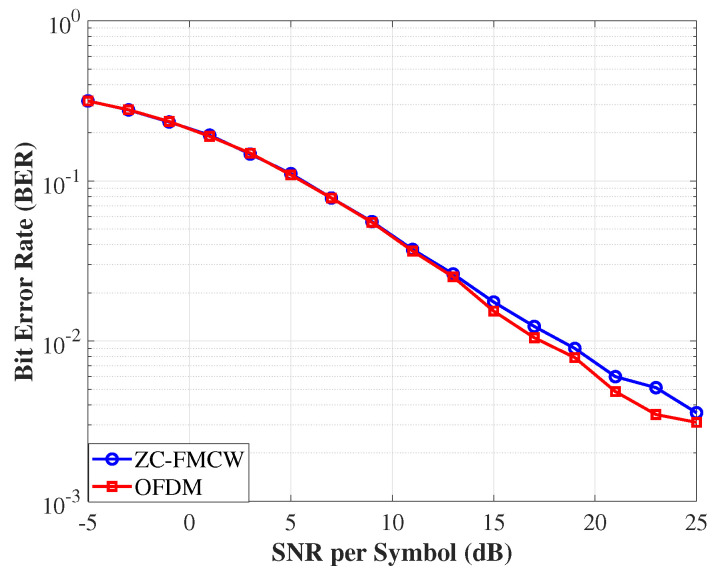
BER comparison between ZC-coded FMCW (blue circles) and multi-carrier OFDM (red squares) under a compound K fading channel. Both schemes adopt 4-QAM modulation and transmit $10^{5}$ bits per SNR level, with BER values averaged over $N_{\mathrm{MC}}=1000$ Monte Carlo trials. The BER curves span SNRs from –5 dB to 25 dB. ZC-FMCW exhibits slightly higher BER at high SNRs, with performance gaps typically below $2\times 10^{-3}$, confirming its competitiveness for communication under strong multipath and clutter.

**Figure 12 sensors-25-05403-f012:**
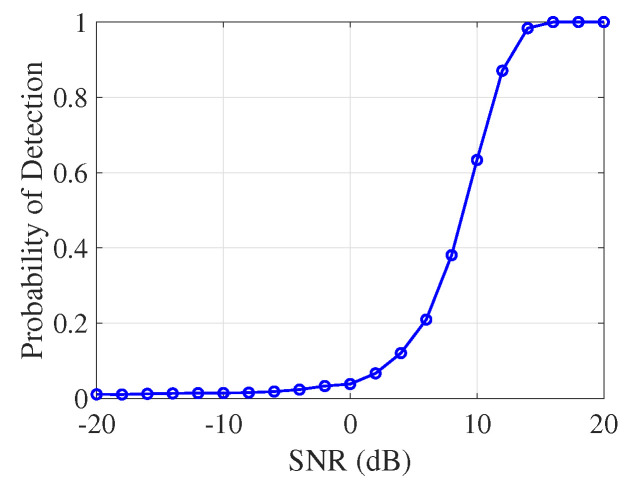
Monte Carlo simulation of detection probability versus SNR for a fixed threshold (γ=3). Each point represents the average probability of detection over 5000 independent trials under compound K-distribution sea clutter.

**Figure 13 sensors-25-05403-f013:**
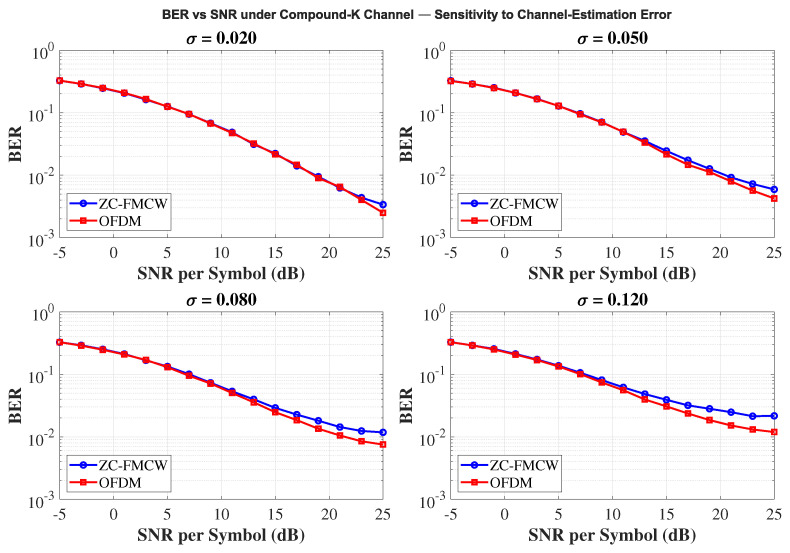
BER vs. SNR under the compound-K fading (m=3, ω=1) for different channel-estimation perturbations σ∈{0.02,0.05,0.08,0.12}. Both ZC-FMCW and OFDM use QPSK and the same MMSE equalizer; For each σ the two waveforms share the same bit and symbol realizations. BER increases monotonically with σ and the high-SNR floor rises, which is accentuated by heavy-tailed compound-K fading.

**Figure 14 sensors-25-05403-f014:**
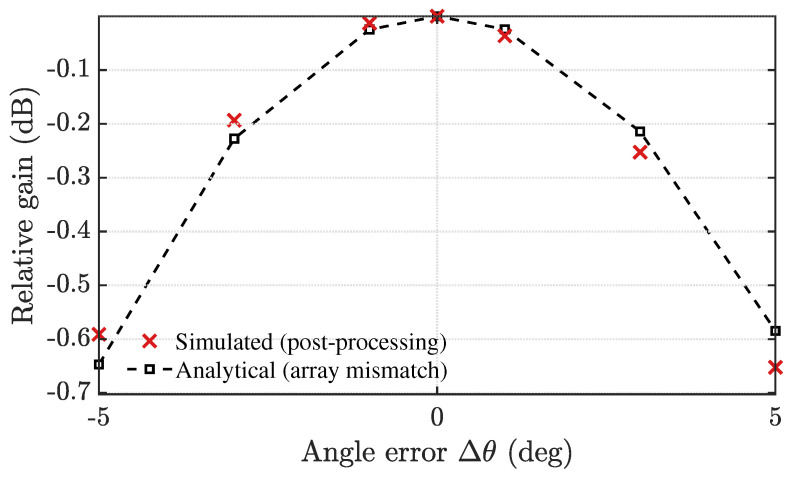
Relative gain versus DoA error Δθ for a 4×4 virtual ULA (element spacing λ/2), normalized to the case that the DoA is accurately estimated, $\Delta \theta =0^{\circ }$. The loss increases smoothly with |Δθ| and remains small within $\pm 3^{\circ }$ (about 0.22 dB), reaching roughly 0.6 dB at $\pm 5^{\circ }$. The results based on the analytical calculation and the system simulation are compared.

**Figure 15 sensors-25-05403-f015:**
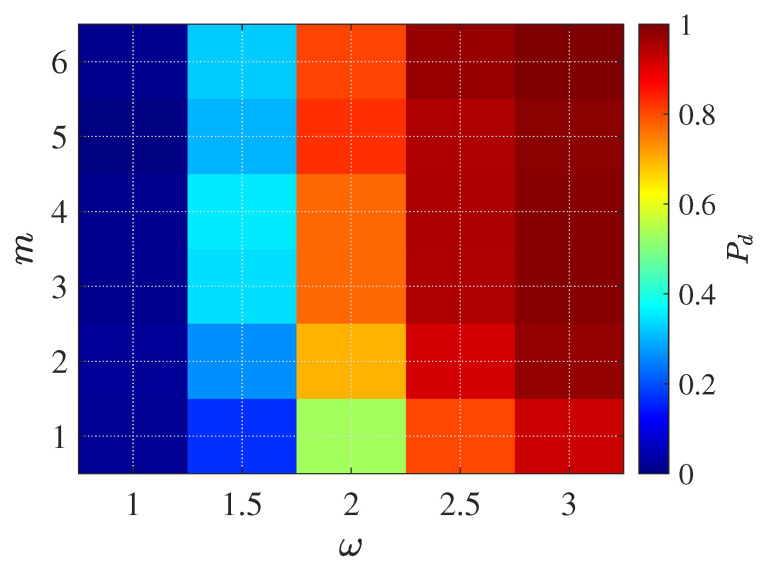
Detection probability $P_{d}$ at SNR = 10 dB as a function of the compound K-distribution shape parameter *m* and scale parameter ω. Each cell shows the Monte Carlo–averaged $P_{d}$ over 500 independent trials, demonstrating that detection performance improves with increasing *m* (less heavy-tailed clutter) and larger ω, and approaches unity when m≥5 and ω≥2.0.

**Table 1 sensors-25-05403-t001:** Comparison of representative maritime SAR and ISAC systems.

System/Method	Δ*R* (m)	Limitation
X-band SAR [4]	1–2	Long revisit cycle; no real-time monitoring.
Marine FMCW Radar [5]	∼0.6	No Doppler or comm.; poor clutter handling.
OFDM-ISAC [6]	∼1	Sensing degrades under heavy clutter.
EO/IR UAV Imaging [7]	-	Visibility-dependent; limited by light conditions.

**Table 2 sensors-25-05403-t002:** Characteristics of code sequences used in autocorrelation over one CPI ($N_{\mathrm{chirps}}=100$).

Signal Type	Base Pattern	Code Length	Total Length
		(Chirps)	(⌈100/CodeLen⌉)
Classic FMCW	-	1	100
Barker	Barker code	13	8×, truncate to 100
Frank (*M* = 4)	Order-4 Frank code	16	7×, truncate to 100
ZC (*u* = 1)	ZC sequence	31	4×, truncate to 100
Costas (*M* = 4)	Order-4 Costas code	16	7×, truncate to 100

**Table 3 sensors-25-05403-t003:** Autocorrelation metrics for the comparison of phase coding schemes.

Coding Scheme	PSLR (dB)	ISLR (dB)
Classic FMCW	0.09	18.17
Barker	1.21	6.70
Frank	1.51	10.04
Costas	1.51	13.44
**ZC **	**3.22 **	**1.35 **

**Table 4 sensors-25-05403-t004:** Settings of the key parameters.

Parameter	Value	Description
$f_{c}$	5GHz	Carrier frequency
$\lambda =\frac{c}{f_{c}}$	0.06m	Wavelength
*B*	150MHz	FMCW bandwidth
$\Delta R=\frac{c}{2B}$	1m	Range resolution
$R_{\max}$	500m	Maximum unambiguous range
$T_{\mathrm{chirp}}$	0.01s	Chirp duration
$f_{s}$	100kHz	Sampling rate
$N_{s}$	1000	Samples per chirp
PRF	100Hz	Pulse repetition frequency
$N_{\mathrm{chirps}}$	1000	Number of chirps per frame
*M*	4	Number of transmit antennas
*N*	4	Number of receive antennas
*d*	0.03m	Element spacing (λ/2)
w	Adaptive	Beamforming vector (MVDR)
Code	ZC	Phase coding
$R_{0}$	50 m	Initial target range
$A_{\mathrm{bod}}$	0.3 m	Torso oscillation amplitude
$f_{\mathrm{bod}}$	0.2 Hz	Torso oscillation frequency
$A_{\mathrm{arm}}$	0.5 m	Arm oscillation amplitude
$f_{\mathrm{arm}}$	1.0 Hz	Arm oscillation frequency
$\sigma_{\mathrm{torso}}$	1.0 m^2^	Torso radar cross-section
$\sigma_{\mathrm{arm}}$	0.9 m^2^	Arm radar cross-section
*m*	3	Clutter shape parameter
ω	1	Clutter scale parameter
$N_{\mathrm{clutter}}$	50	Number of clutter scatterers

## Data Availability

The data presented in this study are available on request from the corresponding author.

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
