# Peer review of "A Phase-Coded FMCW-Based Integrated Sensing and Communication System Design for Maritime Search and Rescue"

_sensors, 2025, doi:10.3390/s25175403_

Round 1

Reviewer 1 Report

Comments and Suggestions for Authors

The paper proposes an ISAC system using phase-coded FMCW waveforms in a 4x4 MIMO architecture for maritime search and rescue. The following changes are recommended to improve the quality of the presented work.

  1. Page 6 line 2: Typo.
  2. Monto Carlo should be corrected to Monte Carlo - Abstract.
  3. Please use consistent terminology: S&R or SAR.
  4. Radiocommunication is two separate words, i.e. radio communication.
  5. How would the performance be affected in the case of bulk Doppler expected for dynamic targets?
  6. What false alarm rate is used for the probability of detection plots?
  7. The caption of Fig. 10 shows 200 trials, however, the text says 1000 trials. 
  8. Can you comment on the effect of sea state on the performance of ISAC?
  9. How would the performance be affected in a real maritime environment, where, e.g. there are occlusions? 

Reviewer 2 Report

Comments and Suggestions for Authors

This manuscript presents a comprehensive study on a 4×4 MIMO phase-coded FMCW-based ISAC system tailored for maritime search and rescue (SAR). By incorporating Zadoff–Chu (ZC) coding into FMCW chirps and modeling sea clutter via a compound K-distribution, the authors address simultaneous high-resolution sensing and reliable communication under challenging maritime conditions. The topic is timely and relevant, especially in the context of next-generation maritime radar systems integrating sensing and communications. The work is generally well-motivated, mathematically detailed, and supported by extensive simulations. However, while the technical approach is solid, several aspects require clarification, refinement, and deeper analysis before the paper is suitable for publication.

  1. While the integration of ZC-coded FMCW for ISAC in maritime SAR is interesting, the novelty relative to prior radar-centric ISAC works is not fully emphasized. In addition, why ZC is applied instead of Flage sequency [R1], which should be discussed in this work. [R1]"Flag Sequence Set Design for Low-Complexity Delay-Doppler Estimation," in IEEE Transactions on Vehicular Technology, 2025.
  2. The proposed contributions are primarily simulation-based; the paper would benefit from a stronger statement on what is fundamentally new in waveform design or detection methodology beyond applying known ZC codes in a maritime clutter context.
  3. The paper assumes static clutter statistics and fixed target positions in most scenarios. Real SAR conditions involve time-varying sea states, moving platforms, and non-stationary clutter, which could degrade performance.
  4. The simulation does not incorporate Doppler spread due to platform motion, hardware impairments, or multipath from ship superstructures, which may significantly affect micro-Doppler extraction.
  5. The energy-based peak selection may not be optimal in dense multitarget or fluctuating SNR scenarios. More advanced detection methods (e.g., CFAR, STAP, sparse recovery) could be considered or at least discussed.
  6. Channel estimation errors are fixed at σest = 0.05; the sensitivity of BER performance to estimation accuracy is not analyzed.
  7. Beamforming assumes perfect knowledge of the target angle; robustness to angle estimation errors should be addressed.
  8. Real-time computational complexity and memory requirements of the proposed processing chain are not discussed, which is important for SAR deployment.
  9. While OFDM is used for communication benchmarking, the paper would benefit from comparing sensing performance against other ISAC waveforms (e.g., OTFS and AFDM). [R2]"A Unifying View of OTFS and Its Many Variants," in IEEE Communications Surveys & Tutorials, 2025.

Round 2

Reviewer 2 Report

Comments and Suggestions for Authors

I have no further comments and suggest accepting the current version.